# Antilogies in Ancient Athens: An Inventory and Appraisal

Livio Rossetti 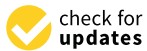

Department of Human Sciences and Development, University of Perugia, 06123 Perugia, Italy;
livio.rossetti@gmail.com

**Abstract:** Antilogies, or pairs of symmetrically opposed speeches or arguments, were generally ignored by Plato, Isocrates, Aristotle, Cicero, and Diogenes Laertius, and, later, by Eduard Norden, Hermann Diels, and most modern scholars of antiquity. As a consequence, until the end of the twentieth century CE, antilogies have been ignored or, at best, treated as a minor literary device to be mentioned only with reference to individual writings. Nevertheless, during the second half of the fifth century, antilogies were a crucially important form of argument and persuasion in 'sophistic' thought, philosophy, historiography, comedy and tragedy, and other fields. In order to redress the historical neglect of the art of antilogy, this essay provides an inventory (doubtless incomplete) of some 30 antilogies composed by playwrights such as Sophocles, Euripides, and Aristophanes, historians such as Herodotus and Thucydides, and, most importantly, 'sophists' such as Protagoras, Gorgias, Prodicus and Antiphon (in addition to a few other writers of the same period). Building on this inventory, the second part of the essay seeks to establish identifying features of antilogy and assess its cultural significance in the Athenian context (in the second half of the fifth century BCE).

**Keywords:** antilogy; sophists; Socrates; Protagoras; Gorgias; Prodicus; Herodotus; Thucydides; Sophocles; Euripides; Aristophanes; Antisthenes; Ancient Greek literature

## 1. Introductory Remarks

It is well known that Athens and Attica knew a unique period of prosperity and creativity in the fifth century BCE, whose fruits became apparent during its second half. The many-faceted expansion of Athens included the exploitation of a completely new sort of communication scheme: couples of opposed speeches aimed not at prevailing over a given adversary or supporting a political proposal or a sentence but, rather, at leaving audiences as puzzled as possible. My main claim in this essay is that these antilogies (*antilogiai*), or legal sets of contradictory speeches, deserve to be counted as one of the most significant accomplishments of Athens in this period. Antilogy originated in Syracuse and Abdera, it seems, but it was mostly in Athens that it knew a first-order flowering during the second half of the fifth century BCE. Probably because of the silence of celebrated authors such as Plato,[1] Isocrates and Aristotle, then Cicero and Diogenes Laertius—a silence continued in the modern period by Eduard Norden in his influential *Antike Kunstprosa* ([Norden 1898](#)) and Hermann Diels in his even more influential *Fragmente der Vorsokratiker* ([Diels 1903](#))—antilogies have long been ignored or treated as a minor typology to be mentioned only with reference to individual writings and authors (for example, Antiphon).[2]

However, evidence exists of some 30 different antilogies authored by 'sophists'[3] such as Protagoras, Gorgias, Antiphon, and Prodicus, playwrights such as Sophocles, Euripides and Aristophanes, historians such as Herodotus and Thucydides, and Antisthenes (best known as a Socratic) and the anonymous author of the *Dissoi Logoi*. Moreover, some of these antilogies or pairs of opposed speeches are highly creative and meticulously constructed. It is therefore important to acknowledge their identity and devote greater scholarly attention to these important types of argument. And since no inventory is available, at least as far as I know, the main aim of this essay will be to offer an inventory of the extant antilogies,

all ascribed to the second half of the fifth century BCE. After this inventory of *antilogiai* composed by 'sophists', rhetoricians, historiographers, philosophers, and playwrights, the second part of this essay identifies the essential properties of antilogies and discusses the cultural context in ancient Athens that gave rise to the sudden efflorescence of this particular art.

## 2. A Tentative Inventory of Antilogies

### 2.1. Antilogies by 'Sophists'

#### 2.1.1. Tisias

The story of antilogy (and of rhetoric) appears to begin with Tisias of Syracuse. Although our sources discuss Corax and Tisias as if they were two people, recent scholarship points to the conclusion that "Tisias was a rhetorician also called Corax."[4] Our sources associate Corax and Tisias (or the sole Tisias) with the invention of several arguments from likelihood (*eikos*). It is Aristotle who reports the following with reference to *hē Korakos technē*:

> If the accused is not open to the charge—for instance, if a weakling is tried for violent assault—the counter-argument is that he was not likely to do such a thing. But if he *is* open to the charge—i.e., if he is a strong man—the defense is still that he was not likely to do such a thing, since he could be sure that people would think that he *was* likely to do so. And so with any other charge. (Rhetoric II 24, 1402a17–21; trans. Barnes 1984 with a modification)

The same example surfaces in Plato's *Phaedrus* (273a–c), where the person mentioned is Tisias. The leading idea is that a weak person accused of assault, but unable to produce eyewitnesses (*martures*) as evidence of his innocence, could rely on an argument from implausibility ("Being weak, I could not have committed that assault successfully"). Now suppose, conversely, that a strong person accused of assault by a weak person has no eyewitnesses to discharge him of the accusation. He could argue that he did not commit the assault because, given his obvious strength, it would have been all too easy to mount a plausible accusation against him. This too would have been an argument from implausibility or improbability but, given the circumstances, one could expect it to be believable. Taken together, these contrasting arguments served as the prototype for the *logos amarturos*, an argument made in the absence of eyewitness testimony.[5] Thanks to these and a considerable number of other schemes of argument,[6] Tisias may well have been the first professor of rhetoric known in Greece, the only point of doubt having to do with chronology: Tisias may have been earlier than Gorgias, but how much earlier?

#### 2.1.2. Protagoras

The next important development in the history of antilogical argument concerns a rather famous competition involving the 'sophist' Protagoras of Abdera and one of his pupils. A dispute between Corax and Tisias mentioned by some ancient authors, notably Sextus Empiricus (*Adversus mathematicos* II 96–99), is better known as a dispute between Protagoras and Euathlos thanks to the testimony of Aulus Gellius (*Noctes Atticae* V 10) and especially Apuleius (*Florida* 18). Recently, Michele Corradi (2012, pp. 38–43) described the story as a mere anecdote, but this story is in fact one of the most perfect examples of antilogy that have survived from ancient Greece.

To begin, let us recount the story. We are told that, in order to be allowed to study with the celebrated (and probably expensive) Protagoras, the young Euathlos offered to pay half of the agreed-upon tuition in advance, with the rest to be paid after Euathlos—strengthened by the skills as a logographer, to be acquired under the master—had won his first case. This arrangement was reportedly accepted by Protagoras. However, contrary to expectations, at the end of his training Euathlos did not practice the profession of logography for a considerably long period of time, and he consequently postponed the payment of the balance in accordance with the agreement that he would pay only after he had won his first case. In order to secure payment nonetheless, Protagoras sued (or threatened to sue)

Euathlos on the basis of the following reasoning: "Beware, Euathlos, I'll open a *dikē* against you in order to be paid. Clearly, if convicted, you would have to pay because convicted; however, in case you were acquitted, you would have to pay because acquittal would mean that you have just won your first case" (Apuleius, *Florida* 18: here and above, I offer a rather free rendering of what Apuleius reports). Faced with the threat of a lawsuit by his teacher, Euathlos is said to have objected, "Not a solid option, dear master. Suppose you sue me. If convicted, I'll abstain from paying because I will still have to win my first case and, if acquitted, I'll abstain from paying because I have been acquitted" (§ 18).

In my opinion, this story is so perfect and so brilliant that it hardly could have been a mere expansion of an anecdote. Indeed, here, as in other antilogies, every detail has been wisely adjusted *in order to* give rise to a situation that leaves nobody able to tell who is right. Therefore, it is unlikely that anybody could have mounted such a story without having a clear idea of antilogies.[7] Indeed, without a definite idea of the goal to be achieved, and without making every detail of the story serve this goal,[8] no well-conceived impasse could be attained. If so, whoever authored the *Euathlos* must have already had a very clear idea of what is required for a text to become an antilogy. The next question, therefore, is as follows: Who is likely to have devised such a well-conceived antilogical dispute? The first point to make is that we are dealing with perhaps the most ancient of the antilogies known to us, provided that we leave aside the arguments ascribed to Tisias. Indeed, the story seems to describe a verbal competition (*agōn*). For these reasons, the *Euathlus* deserves to be acknowledged as a paradigmatic example of antilogy.

No direct evidence is available in favor of Protagoras' authorship, but the emphasis on the professionalism of a student of his, who was able to neutralize the argument mounted by his master (notably, about his fees), seems excellent as an encouragement to choose Protagoras as a master, and is compatible with what is reported by Plato in the *Protagoras* (328b–c) about how Protagoras dealt with his tuition fees. Consequently, Protagoras is the most plausible candidate to be the author.[9] In fact, to point to Protagoras as the probable author of the *Euathlos* remains the only viable conjecture. Furthermore, that Protagoras authored other antilogies that are now lost is quite possible. In principle, Protagoras could have even affirmed himself as the father of antilogies, though this is speculation.

### 2.1.3. Antiphon

If the work of Protagoras provides the earliest examples of *antilogiai* in ancient Greece, the work of Antiphon[10] provides some of the most representative examples of antilogical arguments that survive. Antiphon authored three masterly antilogies—known as his *Tetralogies*—that have reached us in complete form.[11] Here the antilogical aim is reinforced by the choice of adding to the conventional couple of speeches two replies, one for each side. Another qualifying feature of these three contradictory stories is that they are, at the same time, excellent *amarturoi logoi*, that is, disputes not (or not adequately) supported by witnesses. To mount an attractive antilogy with no witness (or with possibly irrelevant witnesses) clearly adds importance to the appeal to the mere plausibility of each claim. At the same time, it adds difficulty to the verbal competition. Implicitly, it suggests that only a specialist could skilfully meet such serious difficulties. Another preliminary deals with the brevity of these speeches, as if they were abridged versions of real forensic speeches to be delivered in front of judges.

The first of the *Tetralogies* deals with a physical confrontation between two wealthy, elderly men, and the accuser presumes to have identified who probably committed the crime, practically without witnesses. The verbal competition, once again, is about likelihood and probability—"This is most likely, that is not likely," countered by, "No, on the contrary, *this* is most likely, *that* is not likely." It is interesting to read the anonymous *hupothēsis* that precedes the fourth speech (the second speech of the defendant): "Here I am, he says, me and my misfortune. . . Here I am entrusting myself to my misfortune and the evil of these people" (I 3 (*Hypothesis*); trans. Gagarin 2008). As proof of not having committed the assault, the defendant invokes the fact that he did not leave the house on the night of the crime,

and makes his servants available for torture to test his claim to truth. This *hupothēsis* tries to capture the sense of bewilderment of the defendant, who has the impression of having no argument with sufficient force to persuade the jury. Nevertheless, the first tetralogy presents many compelling arguments in support of both positions. For modern readers, it is surprising to see the defendant's attitude of distrust toward his alibi (a modern notion), but we have to consider how different may have been the probative value of an eyewitness, especially in light of their social status in the *polis* (for example, whether the witness was a citizen or a slave, a man or a woman). Confronted with this, readers and virtual judges may well have had the impression of being unable to go beyond what each disputant alleged.

The second *Tetralogy* deals with the accidental killing of a young spectator during a javelin contest in a gymnasium. Since the persons directly involved (the killer and the victim) were minors, the debate is conducted by their fathers. Here it is interesting to note that the father of the victim begins with an exceptionally short speech in which he claims that the nature of the tragic event is so clear that it does not need to be debated. However, the father of the young killer unexpectedly dares to claim that his son became a killer only because of the rashness of the victim, who recklessly crossed the area where the javelin was directed, and that he is therefore the true victim in the story. His opponent's claim, in addition to the predictable "This is a case whose precise meaning I can hardly understand, and I am even more perplexed how I should explain it to you" (§ 1; trans. Gagarin), is that the defense speech the audience just heard is plausible but not true (*pistoteron ē alethesteron* [§ 4]). From this, the accuser infers that, in case of acquittal, the menace of impiety would weigh upon the judges themselves. A further surprise lies in store for the judges in the fourth speech, in which the second father involved claims that, since the person responsible for the murder has already punished himself, justice has been done and no further punishment is required. What I have just reported should be enough to convey the spectacular and unpredictable movement from speech to speech in this antilogy.

The third of Antiphon's *Tetralogies* is comparable to the second one in that it outlines an accusation (*stasis*) that opens the door to a counter-accusation (*antenklēma*). A quarrel between a young man and an old man endangers the latter's health to the point that the old man dies after a period of time. The young man is then accused of homicide, but he objects that the death was caused by the incompetence of the physician, who was alerted by colleagues that, if treated this way, his patient was going to die. The defendant adds that, were he unjustly condemned to death, the judges would have become "murderers of your righteousness" (Section 2.5; trans. Gagarin). The prosecutor, in turn, argues that his opponent struck the old man with the intention of killing him, and the law makes responsible the person who commits the assault. At this point, the defendant is said to have chosen to go into exile (as the lesser evil). Some of his friends therefore take over the trial and argue that the causal relationship between the beating and subsequent death of the old man is not adequately supported, and that the judges should therefore acquit the accused while entrusting to time the discovery of the real guilty party (presumably the physician). Once more, the antilogy makes issuing a verdict under these conditions deeply problematic. At the same time, thanks to these three legal battles, it becomes clear how helpful a good logographer (legal speechwriter) could be when facing a trial.

### 2.1.4. Prodicus

Another antilogy, authored by Prodicus, is devoted to the choice of Heracles (and is often labeled *Heracles at the Crossroad*). The contents of this chapter surface unexpectedly, and with many details, in Xenophon's *Memorabilia* (II 1.21–34 = 84B2 DK = 34D21 LM). Here Xenophon offers a careful synthesis and paraphrase of a text by Prodicus meant to be part of a book entitled *Horai* (which failed to survive). Being the sole testimony, there is little space to form more precise ideas about how the summary might differ from the original.

Xenophon (and possibly Prodicus) begins by relating how the hero, a young man, sat down and asked himself whether it was more advisable to follow vice or virtue. While he was searching for an answer, Heracles had the impression that he was approached by two

women, one sober, composed and in a white dress, the other buxom, wearing flashy clothes and visibly eager to give an attractive image of herself. The second woman is the first to address him. She claims that Heracles should follow her because she will lead him on the most pleasant and easy road in life (§ 21; trans. Laks and Most 2016): "To my companions I grant ... benefits [coming] from every possible source" (§ 25). When Heracles asks her to identify herself, she replies that her friends call her Happiness, while her enemies call her Vice (§ 26). Xenophon then recounts the argument presented by the first woman. Her argument is that while pleasure is deceptive, satisfaction and the attainment of goals is the consequence of effort and industriousness. For lands produce fruits in abundance if duly cultivated, a body becomes powerful if exercised with toil and sweat, and, likewise, citizens' admiration for one's virtue depends on the goods brought to them (§ 27–28). Are these merely commonsensical statements? Probably not, because this overview, aside from being unprecedented, is clearly required by the logic of the tale.

Once Vice replies that the other woman envisages a long and difficult road, while her own proposal would be easy and short (§ 29), Virtue presents a qualifying argument: that way you will fill yourself with pleasures (indeed, you will "force the pleasures of sex") before you feel the need for them, and likewise you will fill yourself with (possibly expensive) foods and drinks before you are hungry or thirsty (§ 30). On the contrary, she continues, my friends have a sleep that "is more pleasant than that of the idle," and so on. What is more, she argues, if you simply pursue pleasure in life you will not be honored by gods and men and will therefore be deprived of "the most pleasant sound of all, praise of yourself" (§ 31).

The final words by Xenophon in this chapter deserve further attention: "This, he comments, is how Prodicus presented Heracles' education by virtue, except that he adorned the thoughts in even more splendid words than I have done now" (§ 34; trans. Laks and Most 2016). To read such explicit words of admiration for a writer on the part of another writer of value is very rare, for the time. However, this sentence raises some doubts as to the possibility of envisaging an antilogy in a tale where Heracles is clearly expected to opt for Virtue, or rather for toil and hardship (*ponos*). However, the impression of not being confronted with a 'true' antilogy is probably wrong because the tale is not concentrated on how Arete educated Heracles but on the contraposition of two ways of life before a young Heracles who still had to take a decision. Therefore, the core of this original and living antilogy is not what Heracles will do, but the competition itself, that is, the previously unknown radicalization of the opposition between what could be called, respectively, pleasure and an ordered life. Prodicus' tale is clearly paradigmatic and reaches an uncommon level of universality. It shows how far an antilogy is capable of going.

Indeed, a paradigmatic choice between two very characterized ways of life emerges. While Prodicus seems to have radicalized the alternative between virtue and vice as nobody else before him, other masters (Plato first of all) are known for having spent considerable energy precisely in order to avoid such a choice. They were aware of how unattractive a virtue deprived of every pleasure and a pleasure attained at the expense of social prestige might be. Another feature that qualifies this tale as an antilogy is its choice to leave aside judicial matters, themes suitable to be portrayed in a theatre, and all reference to truth and lies. The functionality of these choices is a qualifying mark of its antilogical features.

### 2.1.5. Gorgias

Gorgias of Leontini is known above all for three highly creative writings: *Encomium of Helen*, *Palamedes* and *On Not Being, or On Nature* (*Peri tou mē ontos ē peri physeōs*, often abbreviated as *PTMO*). The first two of these texts are still available, while the *PTMO* is known only thanks to two excellent summaries, one in Sextus Empiricus' *Adversus mathematicos* and the other in the *Corpus Aristotelicum*.[12] Each of these works by Gorgias offers arguments in support of a claim that is opposed to a virtual (never written) anti-*logos* that, nevertheless, is readily discernible by readers.

At first sight, the antilogies ascribed to Gorgias may seem, in comparison to others, rather conventional, at least when considering his encomium of Helen in light of the current, traditional portrayal of her demerits, as well as his defense of Palamedes against the accusations mounted by Odysseus. This, however, is only a superficial side of the story, as will be shown.

What is immediately evident in the *Encomium of Helen* is its thematic richness. It is not by chance that scholars have devoted considerable attention to Gorgias' ideas about play and game (*paignion*), love and seduction, pleasure and persuasion, poetry and prose, *kairos*, truth, as well as his theories of speech and use of definition, his idea of persuasive speech as *pharmakon*, his *gorgiazein*, and other points of detail.[13] far less attention has been devoted, so far, to the most important feature of this epideictic speech: its systematic treatment of the problem of the limits of the will (and therefore of the limits of freedom and responsibility), that is, to the ways in which one's behavior may be conditioned by forces that lie beyond one's control. In fact, the whole *Encomium* deals with this topic, and nobody before or after Gorgias is known to have dealt with the limits of the will so systematically until Martin Luther addressed this topic in his 1525 *On the Bondage of the Will* (*De servo arbitrio*).

Since I have argued this point elsewhere,[14] here it may be enough to remark that Gorgias deals in a systematic (but general) fashion with four possible sorts of conditions that may have affected Helen's decision to leave Sparta to go to Troy with Paris: gods, force, persuasion, or love. What is more, he devotes the body of his speech to each of the four possible conditions in an orderly fashion, almost as if he were writing a treatise devoted to highlight the limits of the will, and no other side of the story. In fact, Gorgias says nothing about how the pressure exerted to make Helen leave Sparta could have been resisted or about her co-responsibility for the decision.

These remarks should make clear the intentional one-sidedness of Gorgias' crypto-treatise on the limits of the will, which is designed to counter the traditional accusation that Helen was the sole cause of the Trojan War. *Prima facie*, with the *Encomium of Helen*, Gorgias aims to refute the widespread idea that Helen bore sole responsibility for the Trojan War, but this is merely the surface of the text. Behind it lies a well-structured treatise designed to account for the force that gods, fate, persuasion, or love can exert upon what we would now call 'the agent'. Therefore, the many themes mentioned at the beginning of this section (*paignion*, *kairos*, *pharmakon*, etc.) serve as mere ingredients in a rich argument whose real achievement is to offer, in a slightly camouflaged form, the first treatise on the limits of the will in Western culture. What is new—and remarkable—here is the quiet coexistence of surface and subtext, much as if the author wanted to be appreciated both for a fluent surface, suitable to please a rather superficial audience, and a subtext suitable to gratify a more penetrating group of potential listeners and readers. Behind the treatise, there is the well-established penchant of most Greeks to decline one's own responsibility by appealing to fate or divine intervention. Indeed, if Homer's main heroes can rely on divine assistance whenever they are in danger, the intervention of a god, or, more often, Fate, continued to be invoked in a number of fourth century judicial orations, much as if this assumption was still commonly accepted. What is also new in *Helen* is the attention paid to love and, to a greater degree, language and persuasion. It is not by chance that many scholars judge the section on persuasion so important that they identify it as the key topic dealt with in *Helen*. As a matter of fact, the topic of persuasion is clearly the most original, but there is little doubt that it remains but one element of a more comprehensive whole. We cannot ignore that Gorgias was explicit in giving the same list of vices of the will at § 6 (towards the beginning, after some preliminaries) and § 20, when he is about to conclude his speech.

Another fully developed antilogical work by Gorgias is his *Palamedes*. In comparison with the *Encomium of Helen*, the one-sidedness of this forensic speech is even more evident, since it is designed to counter an accusation speech supposedly delivered by Odysseus in front of the same jury, and while the siege of Troy was continuing. The main architecture of Palamedes' defense consists in a double claim: (A) "I would not have been able to betray the Greeks even if I had the *intention* to do so," and (B) "I could not have wanted to betray

them even if I had had the *opportunity* to do so"[15] With this arrangement, Gorgias is able to identify and refute, one after another, a number of preconditions, opportunities, hindrances, and individual behaviors that would need to have happened to tempt Palamedes to betray the Greeks, as well as a set of subjective conditions the speaker judges to be necessary for the decision to betray Greece to be taken (§ 6–21). The material impossibility of the supposed betrayal is one of the points argued in detail:

> "How could I have brought them [the Trojans] in? Through the gates? But it is not up to me either to close these or to open them, but it is the leaders who are in charge of these. Or over the fortifications with a ladder? Not at all. For they are all full of guards. Or by making a breach in the walls? Then this would have been visible to all. For life under arms . . . takes place in the open air, in which <all men> see all and all men are seen by all". (§ 13 = 82B1 DK = 32D25 LM; trans. Laks and Most 2016)

The order presiding over both sets of denials makes the force of the overall argument apparent. Based on these premises, Palamedes continues his speech by arguing that Odysseus' accusation bears, moreover, the mark of incompetence (he failed, for example, to consider all that has been analytically examined in the speech [§ 22]), and he goes on to add that he (Palamedes) was portrayed by Odysseus as being simultaneously cunning (because involved in a sophisticated scheme to betray his fellow Greeks) and stupid, if not insane, since he would have acted to his own detriment. But to be both cunning and stupid at the same time is manifestly impossible, since these qualities are patently contradictory. Therefore, Palamedes concludes, it is Odysseus who is clearly unreliable in his accusation (§ 25–26).[16]

Having shown that (a) he could not have betrayed the Greek army (lack of objective conditions), (b) that he could not have the least interest in doing so (lack of subjective conditions), and (c) that the prosecutor patently contradicts himself, Palamedes adduces a number of further arguments, namely that (d) he has never been the subject of the least complaint, (e) he has been a great benefactor of mankind, (f) his character is only commendable, and (g), since the judges ("you, who both are and are reputed to be the first of the Greeks" [§ 33]) have an evident interest in avoiding mistakes, he can confidently expect them not to make a mistake in an irremediable matter such as this one (his possible condemnation to death as a traitor).

The supreme mastery of the *Palamedes* makes it a model, probably unparalleled, of how to get around the difficulty of providing direct evidence to establish a fact. It consciously relies on the sole force of reasoning based on common sense, without any support from eyewitnesses. The *Palamedes* is therefore a dizzying *amarturos logos* in which reasons are capable of convincing despite the absence of supposedly 'objective' proof. In this epideictic speech, each element shows an uncommon degree of functionality from the point of view of its overall (architectonic) project. Precisely because of the high level of functionality of each part, a truly argumentative vertigo is attained, and it is easy to grasp how demanding it must have been to reach excellence in making Palamedes' self-defense such a superb refutation.

We come now to the most virtuosic of Gorgias' works known to us, his *On Not Being, or On Nature* (*Peri tou mē ontos ē peri physeōs*). The original text has not survived, but good luck has preserved two detailed, competent and surprisingly complementary summaries.[17] One is by an anonymous author and is erroneously included in the *Corpus Aristotelicum*; the other, by Sextus Empiricus, is included in his *Adversus mathematicos* (book VII). Thanks to these two first-order synopses of *Peri tou mē ontos ē peri physeōs* (commonly labeled *PTMO*), we are able to form a relatively clear idea of what Gorgias is likely to have argued in this work.

Gorgias' book is indisputably paradoxical even in its title. In the background is Melissus' *Peri phuseos e peri tou ontos*, but while Melissus' book was in all likelihood centered on the notion of being (*phuseos*), as its title indicates, Gorgias, starting from the title, dared to credit a totally absurd idea, namely that his was a book dealing with "what

does not exist".[18] As a consequence, people who received a copy of it, or were invited to buy it, could only ask if Gorgias was *really* going to argue that nature does not exist. A look at the contents of this book can only confirm the impression of bewilderment. As is known, his treatise is explicitly governed by the following series of bizarre claims: nothing exists; supposing that my first argument fails, you will concede at least this, that nothing of what exists can be known; and supposing that even my second claim is unconvincing, you will acknowledge at least this, that nothing of what exists and is known can be communicated. Gorgias is clearly mounting an attack on the basic tenets of common sense. For this reason, we are entitled to see in it, not unlike *Helen*, an argument facing a powerful adversary: common sense. A virtual *logos* is therefore clearly contrasted by Gorgias's anti-*logos*. Indeed, his *PTMO* pretends to contradict common sense in the most global way.

With its three main theses—"nothing exists," "nothing can be known," and "nothing can be communicated"—and a host of supporting arguments (some of them very technical), Gorgias' *PTMO* mounts an attack on common sense that is an amazing exhibition of boldness, a tremendous *tour de force* totally indifferent to the patent unattainability of its stated goals. One is reminded of Gottfried Leibniz's claim that each of us is a monad closed in on itself. But if we are totally closed in on ourselves, how could we know something that is totally exterior to us? To realize that Gorgias went so far more that two thousand years before Leibniz, when philosophy was still little more than a word, is amazing. The bold claims of Gorgias' *Helen* and *Palamedes* pale in comparison to the visionary features of the *PTMO*[19].

To enter into more details about Gorgias' texts is beyond the scope of this survey of *antilogiai* in the fifth century BCE, but a quote is nevertheless appropriate:

> Even if speech exists, he says, it nonetheless differs from all the other things that exist, and there is nothing that differs more than visible bodies and speeches. For what is visible is grasped by one organ, speech by a different one. So speech does not indicate the multitude of things that exist, just as these do not reveal their nature to each other. (Sextus Empiricus, *Adversus mathematicos* VII 86 = 82B3 DK = 32D26b LM; trans. Laks and Most 2016)

These are the last words of the summary by Sextus Empiricus, whose contents are largely confirmed by the summary in the *Corpus Aristotelicum*. It may be pertinent to conclude this section by asking whether from these summaries—or from a more comprehensive set of testimonies—we can extract a definite idea of what Gorgias might have taught, in the event he wanted to teach anything. The key point, however, is that even if some hints appear, he did nothing to single out his real tenets and make them known to us in a clear fashion.

## 2.2. Antilogies in Historiography

### 2.2.1. Herodotus

The new literary fashion of composing pairs of antithetical speeches did not go unnoticed by other writers in the fifth century BCE, and the great historians of the time, Herodotus and Thucydides, did not fail to include some significant example of antilogy in their works. For instance, Herodotus included in his *magnum opus*, *The Persian Wars*, the celebrated *tripolitikos logos* (III 80–82) or threefold political speech in which someone pleaded for democracy, someone else for oligarchy and a third one for monarchy. Before mounting a triple *logos*, Herodotus prepares a very functional context for it: he tells the spectacular victory of the Persians over the Magoi, with mention of the annual celebration of this event, when Magoi were (or felt) confined to their homes. A few days later, the story continues, the architects of the victory conferred about the situation as a whole. One of them, Otanes, launched the idea of giving power to all Persians.

> "It is my view," he says, "that we should put an end to the system whereby one of us is the sole ruler. Monarchy is neither an attractive nor a noble institution. You have seen how vicious Cambyses became and you have also experienced

similar behaviour from the Magus. How can monarchy be an orderly affair, when a monarch has the licence to do whatever he wants, without being accountable to anyone?... What about majority rule on the other hand? In the first place, it has the best of all names to describe it—equality before the law [in Greek: *isonomia*]. In the second place ...". (Herodotus, The Histories III 80.3, 6; trans. Waterfield 1998)

Then Megabizos, the second speaker, famously argues in favor of an oligarchic regime, since it would be foolish "to escape the arrogance of a tyrant to fall into the arrogance of a multitude" (81.2), while "it is logical that the best decisions come from the best men" (81.3). Finally, the third speaker, Darius, argued that "nothing can be preferable to one man in power, if he is the best" (82.2), while in an oligarchy "everyone wants to excel, the oligarchs hate each other, hatred gives rise to hatred, massacres arise from sedition and from massacres one moves on to the government of a single" (82.3).[20]

In these exchanges, when pleading for one of the conceivable options, everyone concentrates on the drawbacks of just *one* of the others. As a consequence, the door for further steps of the exchange (on the part of whichever audience or readership) remains open. This is typical of the antilogies, whose aim was not to teach or persuade but to generate a stubborn perplexity. Let me also recall that no previous debate on the main 'constitutional' options is known, either in Greek or in other languages, and even the corresponding Greek words (*monarchia*, *oligarchia*, *dēmokratia*) started to get some circulation at least in Athens (in the same period), thanks to Herodotus, though not solely thanks to him.

### 2.2.2. Thucydides

Thucydides also enjoyed crafting a couple of unmistakeable antilogies in his *magnum opus*, *The Peloponnesian Wars.* Justly famous is his "Dialogue on the Mytilenians" (III 37–49). The Mytilenian dialogue is supposed to have occurred in the Athenian Assembly when, once the order to exterminate the Mytilenians was decided upon and transmitted (by means of a trireme) to the Athenian troops taking control over the island, citizens and magistrates were prompted by the extreme severity of the punishment to re-examine the question. This time, it was Cleon, the controversial leader indirectly but severely portrayed by Aristophanes in his *Wasps*, who spoke first in support of the decision taken two days earlier to exterminate the Mytilenians, while Diodotos took the floor to plead for the Athenians to reconsider the order. According to the Thucydidean report, Cleon argued that although the Athenians failed to "reflect that your empire is a tyranny" (3.37.2), the Mytilenians chose ''to attack us at what they thought would be the moment of their advantage, not for any wrong done them by us... But as things are now they must be punished in a way which fits their crime... if you lay claim to continued rule irrespective of propriety, then it follows that you should punish them in your own interests too, and forget about equity—or else abandon your empire and make your noble pretenses when nobility is no risk... Punish them as they deserve, and set a clear example to the rest of the allies that the penalty for revolt will be death" (Thucydides, The Peloponnesian Wars, III 39.3, 39.6, 40.4, 40.7; trans. Hammond 2009).

To such a well-conceived argument Diodotos is said to have opposed arguments that are, unfortunately, too sophisticated to be summarized here in a comparably brief way. But it is enough to note that, although arguing against such a severe treatment of the Mytilenians, Diodotos acknowledged the imperialistic attitude of Athens. In the sequel, we are told that Diodotos' proposals were followed almost immediately by a vote in which the decision to spare the Mytilenian citizens narrowly prevailed over Cleon's proposal. As a consequence, another trireme was immediately dispatched to the island, and the sailors were asked to do their best to overtake the first trireme to avoid arriving too late to prevent the slaughter (they arrived in time to countermand the order). That here the contraposition of two opposite ways of understanding what the situation required reaches admirable standards is commonly acknowledged, all the more since Thucydides himself remarks that

here the weight of one point of view is perfectly balanced by the other point of view (III 49): no statement could have been more explicit.

Like the Mytilenian debate, the Melian dialogue is another important example of the use of antilogy, but this time we are confronted with an unexpected exchange—a sort of direct dialogue. According to Thucydides, Athenian generals (*stratēgoi*) resolved to send ambassadors to Melos before starting the looting of the island and charged them with opening the avenue to direct verbal exchanges with the Melians (Thucydides, *The Peloponnesian Wars*, V 85). The Melians immediately argued that their counterpart was both interlocutor and judge, and that the only options left to them were, therefore, either war or enslavement (V 86). After some further steps, the Athenian delegates claim that "we want to exercise supremacy over you by guaranteeing your safety, which would be convenient for us and for you" (V 91), to which the Melian representatives object, "How could it be equally convenient for us to be slaves and for you to dominate?" (V 92). After these powerful preliminary exchanges, the Athenian delegates continue by stating what, in their opinion, would be advisable for the Melians to do, while the Melian delegates do their best to state what would be advisable *for the Athenians* to do. After further exchanges (§ 93–111), the Athenians are reported to have taken the decision to seize Melos and then, some months later, to have destroyed it. This superb dialogue is commonly taken to be a fine example of antilogy and one of the apices of Thucydides' history writing.

*2.3. Some Further Antilogies: Antisthenes' Ajax and Odysseus, and the Dissoi Logoi*

2.3.1. Antisthenes

One of the most elderly followers of Socrates, Antisthenes, authored a couple of speeches, titled *Ajax* and *Odysseus*, that have reached us (*SSR* V A 53–54). These antilogies have nothing to do with his well-known tenets about language, behavior and, more generally, philosophy. For this reason, they are likely to have been the work of a young Antisthenes who was not yet attracted by Socrates, and designed to compete with the best antilogies of the time. In these contrasting arguments, Antisthenes gives voice to the famous dispute over the armor of Achilles: Which of these two heroes, Ajax or Odysseus, would have deserved the armor? And why?

The speech of Ajax opposes his victory in the great fight around the body of Achilles to an outrageous act on the part of Odysseus—the theft of the image of the goddess from Troy—and suggests that his competitor was impious in addition to being inferior as a warrior. Indeed, he could only use the armor in the most worthy way, whereas Odysseus could only sell it, since he knows that, if he wore the armor, his cowardice would become manifest (§ 3). However, the point Ajax stresses is another, quite unexpected one, namely that, much as no good doctor would ask other doctors to diagnose a disease, no king should turn to other people to judge, since judges can only judge on what they do not know (§ 4 and 1). The relevance of the latter claim becomes clear when one considers that in Athens the popular *dikastēria* were comprised of generally incompetent judges who neither requested information nor received information from the president of the *dikastērion*, so that they could ignore both the facts to be judged and the laws relevant to the case (in fact, the momentous task of supplying the basic information was normally left, at least in classical Athens, to accusers and defendants[21]). For these reasons, Ajax's warning about the risks associated with the ignorance of the dicasts could well have sounded attractive in Antisthenes' time.[22]

Adhering to an implicit rule of the antilogies, the anti-*logos* by Odysseus is rooted in quite different arguments, so the more significant arguments endorsed by Ajax remain without response. Some of the arguments Odysseus opposes may be taken to be introductory. One rather weak argument consists of arguing that, "If you hadn't saved Achilles' body, two others would have" (§ 11); another consists of claiming, "When they say you're strong, you're happy like a child" (§ 7); yet, another claims, "Don't you know that the wisdom about war (*sophia peri polemos*) and being distinguished by *andreia* are not the same thing?" (§ 13). In a sense, these are mere premises for Odysseus to ask, on one hand,

"What's the point of calling the man who recovered the statuette of the goddess sacrilegious and not Alexander, who took advantage of us Greeks?" (§ 3), and, on the other, to argue that "If it was noble to take Ilium, it is also noble to find a way to do it" (§ 4). A further counter-argument claims that "while the battles of all of us against all of them proved inconclusive, I ran the peril alone. If successful, it would have been to all; in case of a failure, the damage would have been to one man" (§ 2).[23]

It is interesting to note that Antisthenes seems to have no idea of how arguments could be arranged in a functional order. This is a manifest, though not catastrophic, failure in the arts of persuasion and argument. Gorgias or Antiphon, for example, would never have left things in such disarray. Because of this lack of order, the present antilogy could be an early work, one endowed with originality and force but weakened by an evident compositional flaw.

2.3.2. The *Dissoi Logoi*

Other antilogies come down to us as portions of the so-called *Dissoi Logoi*. Its anonymous author devotes the first chapters to sketching out a small group of antilogical disputations dealing with the topics of "Good and Bad," "Seemly and Unseemly," "Just and Unjust" and "True and False," while departing, step by step, from the genre of antilogic.

The writer begins his first chapter by acknowledging that *hoi philosophountes* (not exactly the philosophers) find it customary to speak of the good and the bad as being different (i.e., distinguishable) things, and immediately states that he sides with those who claim that the same event may well be good for some and bad for others, depending on the circumstances. The author then proceeds to argue that if the good and the bad are the same, a number of absurd consequences will follow. When dealing with seemly and unseemly (second chapter), the writer lists a series of examples to deny that they are the same since the same things (or behaviors) seem beautiful or ugly depending on circumstances, such as the different customs among Lydians and Greeks. The author concludes as follows:

> "As for me, I am astonished that things that were unseemly when they were brought together become seemly and do not remain as they were when arrived. At least if they had brought horses, cows, sheep, or people, they would not have taken away something different; for if they had brought gold, they would not have taken away bronze either, and if they would have brought silver, they would not have carried off lead. So do they take away seemly things instead of unseemly ones? Come then, if someone brought something unseemly, would he lead it away again as seemly?". (II 26–28 = 90.2 DK = 41.2 LM; trans. Laks and Most 2016)

As to the just and unjust (third chapter), the author dares to claim (though not without hesitations) that, under certain circumstances, it is just to do what otherwise would be unjust. However, starting with the fourth and fifth chapters, the author stops mounting contrasting arguments and prefers to argues that these contrasting arguments (*dissoi logoi*) are untenable. For example, in 4.7 the author offers some interesting observations on the problem of time and reference in discourse: "If the event occurs then the speech is true, while if it does not, then it is false. "Similarly in 5.9 the author pursues the question of time and reference through the concept of *kairos*, or "the "right moment": "When someone asks them, they answer that they are the same things, but the wise says them at the right moment, the insane when it is not right.

Here the writer is now leaving aside the typical conventions of antilogies: and instead declaring his beliefs in a rather direct way offering a number of short remarks and going in multiple directions. Indeed, in the sequel, it is not easy to identify where the author is going, or what precisely he wants to stress. The rest of the booklet, as it has transmitted to us, encompasses a number of different remarks, much as if the whole still had to receive its 'last hand'.

One topic not to be ignored deals with the word *sophistai*. In *Dissoi Logoi* we encounter repeated mentions of this word (it occurs five times). These mentions of *sophistai* seem

crucial as evidence of a time when the category of intellectuals known as 'the Sophists' began to be acknowledged as such.[24]

*2.4. Antilogies in Tragic and Comic Theatre*

2.4.1. Sophocles

Greek tragedies are marked by their portrayal of tension, a conflict of interests or a serious misunderstanding between protagonists. Therefore, the portrayal of the arguments of antagonistic characters opens the way to the inclusion of antilogies in the development of tragic conflict. Consider, for example, Aeschylus' *Oresteia*. The supreme clash of the opinions for and against Orestes occurs in its third play, but in principle, the poet *could* have also portrayed the competing advice of Clytemnestra and Agamemnon as to the destiny of Iphigenia, and that would have been a perfect subject for an antilogy. Another superb antilogy *could* have opposed Orestes to his mother. However, nothing of the sort occurred. It is only in the *Eumenides* that we find rather structured speeches. One is by Orestes (443–488), but then, instead of an anti-*logos:* we find a sustained exchange between the chorus and Apollo (614–673), and when we are presented with some elaborate exchanges between the Erinyes and Athena (778–1047), no opposed speeches in the play can be identified as true *antilogiai*. Why? The antilogies probably remained unknown to Aeschylus.

This is not the case with Sophocles, in whose plays something much more like antilogies appears. To begin with, in his *Philoctetes* we find Neoptolemos firmly rejecting the proposal made by Odysseus, though he ends up accepting it (55–129) and, towards the end, Philoctetes attacking Odysseus (1004–1061). In both passages, the typical tension of the drama affects the exchanges. As a consequence, treating them as clearly marked by antilogical features becomes risky. The *Antigone*, on the contrary, portrays an open clash of opposed speeches three times: between Creon and Antigone (443–525), between Creon and his son Haemon (639–723), and between Tiresias and Creon (988–1086). During the verbal competition between Antigone and Creon, the daughter says, "I don't like any part of your *logos*, nor you of mine" (synthesis of 499–501; trans. mine). After the exchange between Creon and Haemon, the son says, "You want to monologue. You never listen" (757). Such statements, combined as they are with the stubborn persistence of each character in their position despite the structured and explicit invitation to re-examine the state of affairs (705–723), seem to capture in both arguments specific antilogical features. Indeed, taken together, these well-combined contrasting speeches show that Sophocles, unlike Aeschylus, had become aware of the phenomenon of antilogy.

*Oedipus the King* also features *antilogiai*. It is true that here we find no conventional dramatic *agōn*, since speeches are not ranged against each other in direct competition, and a 'truth' is being discovered step by step. Indeed, parricide and incest come to light on the part of persons who, other than being unaware of their primary parental ties, discover and acknowledge all that with the greatest distress, and without the least attempt at denying the truth. However, the reference to the unavailability of eyewitnesses—the *amarturos* feature of the dispute—becomes explicit twice (116 ff. and 293), while at 543 f., Kreon is able to ensure that he will reject Oedipus' claims "point by point."[25] Moreover, the word *antilexai*—clearly derived from *antilegein* ("to contradict")—makes its appearance during the verbal confrontation between Oedipus and Tiresias at 409.

These features suggest that Sophocles had a clear idea of what antilogies are and how they work. We may retain doubts about his wish to offer fully developed antilogies, but he may well have had some interest in pointing out that antilogies were known to him. For these reasons, it remains unclear whether his tragedies include intentional antilogies and how many there are in the plays. But to leave Sophocles outside the specialized world of antilogies would be inappropriate.

### 2.4.2. Euripides

Things change with Euripides, who included some spectacular antilogies in his tragedies. Euripides was deeply familiar with 'sophistic' thought, and it is thus likely that he was aware of the art of antilogy practiced by Protagoras, Gorgias, Prodicus and other 'sophists' and intellectuals. It is especially in *The Trojan Women* that we find a direct verbal confrontation between Hecuba and Helen. After a number of short speeches by Cassandra, Hecuba and Andromache, when Menelaus is now determined to kill Helen, the latter is given the opportunity to argue rather extensively for her own innocence. Hecuba, who is present, significantly pleads for Helen to be allowed to give her own evaluation of the events, though on the condition that she too will be allowed, in turn, to present her own version of the facts (906–908). Menelaus agrees, and Helen gives her version of the events (914–965), which is followed by a reply on the part of Hecuba (969–1028). Each of them examines the whole story from opposite points of view. It is unnecessary to enter into details to conclude that there is enough evidence here to acknowledge the existence of deliberate anti-*logoi*. Indeed, to excerpt this pair of speeches from the tragedy would be enough to have a couple of contrasting speeches suitable to count as self-sufficient examples of antilogical arguments.

Something comparable appears in Euripides' *Hecuba*. After an exchange between Hecuba, Odysseus and Polyxena (251–378), the exchange between Polymestor and Hecuba (1132–1237) displays clear antilogical features when they plead quite opposite conclusions about Hecuba's destiny. The final words by Agamemnon are also noteworthy: "Perhaps killing a guest counts for nothing with you, but with us, the Greeks, it is a disgrace. ... How could I avoid criticism if I acquitted you? I cannot" (1246–1250; trans. mine). These words show that the previous speeches were conceived of as formal defense and accusation speeches, with Agamemnon serving as judge. An impeccable *logos amarturos* features no less clearly in his *Hippolytus*, when Hippolytus, openly accused by Theseus, struggles to prove to his father that, although he has no material evidence to exhibit (while Theseus does), the accusation against him is groundless (936–1035). The chorus immediately comments: "You have presented a sufficient rebuttal of the allegation" (936; trans. mine)—a sentence that means, in effect, "You have been able, at least, to rebalance the accusation by opposing good arguments to those of your father." There is enough to treat this exchange, too, as a conscious adherence to the antilogic scheme. The same holds true for another Euripidean tragedy, the *Orestes*, in which a couple of opposed claims are argued with care. However, here the extensive clash between Orestes and Tindareus (491–629) lacks creativity: only the basic antilogical features appear.

Taken together, these examples from Aeschylus, Sophocles and Euripides demonstrate that antilogical disputation was an important feature of Greek tragedy in the second half of the fifth century BCE.

### 2.4.3. Aristophanes

While antilogical debates were central to Greek tragedy, they also took center stage in Old Comedy, most notably in the plays of Aristophanes. What immediately comes to mind is the well-known competition between just and unjust *logoi* that occurs in the second half of the *Clouds*. What we read here is not a pair of speeches but two steps of a sustained contraposition. After some rancorous verbal exchanges between the just and unjust *logoi* (889–933), the chorus asks both competitors to present their views of *paideia* (traditional and 'modern' education, respectively) to ensure that Pheidippides can decide whether or not to enroll in the Think-Shop (*Phrontistērion*) and study Socrates' unjust, 'sophistic' form of *paideia* (934–938). Once the challenge has been accepted, the chorus comments: "Now the two, relying on very dextrous arguments and thoughts, and sententious maxims, will show which of them shall appear superior in argument" (949–952; trans. mine). What the audience expects is a pair of long, opposing speeches between *Dikaios logos* and *Adikos logos* in the form of standard antilogies. What is offered instead is an extended presentation by *Dikaios logos* (961–1023), with two short interruptions by *Adikos logos*, followed by a counter-

speech by his competitor that is, in turn, interrupted repeatedly. Clearly, the interruptions do not affect Aristophanes' adherence to the antilogical model, and the *agōn* between the two *logoi* in fact reveals the playwright's mastery of the genre.

While nothing like this verbal *agōn* appears in *Acharnians* or *Knights* (Aristophanes' first award-winning plays, in 425 and 424 BCE, respectively), one year after the *Clouds*, in the *Wasps* (422 BCE), Aristophanes stages a parody of two opposed *logoi* to be presented before the *dikastai*. To begin with, Bdelikleon says (891), "If there is any judge nearby, let him enter," to which Philokleon replies (893), "Who is the defendant?" Bdelikleon then says, "Here is the *graphē*" (894; trans. mine). Despite such explicit preliminaries, what follows (984–1008) is not a conventional antilogy but rather a parody of what an antilogy could have offered. As such, the contrasting arguments entered here by each speaker serve as conscious substitutes for a formal antilogy.

Comparable to this is the competition between Aeschylus and Euripides in *Frogs*, in which speeches (almost 600 lines: 830–1410) are fragmented because of the continuous exchange of comments from the competitors as well as from Dionysus and the chorus, who also chime in. However, in this debate, both Aeschylus and Euripides carefully analyze their own as well as their competitor's art, and each maintains his own position consistently. What happens in *Frogs* is therefore another extended and highly competent parody of what a formal antilogy could offer.

These remarks sketch out only the most conspicuous aspects of the role played by antilogy in Greek tragedy and comedy. In all likelihood, a more systematic investigation—to be extended, in theory, to a host of other works by these and other Athenian playwrights, which are known only in fragmentary form—would show that, at least in the last three decades of the fifth century BCE, the most celebrated playwrights were quite familiar with the new literary genre of antilogy: their plays reveal an interest in creating an hybrid form that fuses theatrical models and antilogical models. Indeed, the plays offer the best proof of how antilogy permeated the Athenian avant-garde of the period and indicate the success of antilogy as a new form of argument and composition.

*2.5. Some Basic Remarks on this Inventory*

With very few exceptions, all of these texts come to us in complete form. Many of these antilogies by sophists, philosophers, historiographers, and playwrights are well known, and some are even celebrated. We therefore have the unexpected privilege of having direct access to some 30 different antilogies by a considerable number of distinguished authors. Moreover, with just one or two exceptions, all of these writers worked in Athens in the second half of the fifth century BCE—none before, none after, and none elsewhere. One has the impression that most of the prestigious cultural elite of the time—*sophoi*, rhetors, historians and playwrights living in Athens—thought it worthwhile to devise, each in his own way, the most sophisticated antilogies of which they were capable. In addition, many of these texts are highly creative, with those authored by Antiphon, Gorgias and Thucydides among the most inventive and ingenious. And if some of the best writers of the time devoted themselves to writing antilogies for some decades, and more than one—notably Gorgias, Antiphon, and Thucydides—produced their best work in this genre, it is likely that many ordinary Athenian citizens were able to note, understand and appreciate this variety of antilogies.

The present survey raises a question of quantity: How common was antilogy in this epoch? How Protagoras contributed to the birth of this genre remains unknown, but if he authored the *Euathlos*, it is likely that he authored other good examples of antilogy that have been lost. As to the playwrights, although no attempt to investigate the large corpus of fragmentary texts of tragedy and comedy has been made here, to do so would probably add further evidence of the popularity of *antilogiai* in this period. Moreover, leaving aside Tisias and Protagoras, it is quite possible that some emulation of antilogy took place outside Athens, although no evidence of this appears to be available. For all these reasons, the total number of antilogies that actually circulated in the Greek world (irrespective of what we

can still read), and the number of writers involved in their creation, is likely to have been much greater than what the present survey suggests.

What is certain is that, with antilogy we are dealing with, a phenomenon of considerable proportions, that fascinated writers and thinkers from many different fields and professions, took place. The art of antilogy affirmed itself as a significant scheme of literary activity, and some Athenian intellectuals of the period chose to author whole sets of antilogies. The same holds true for some foreign intellectuals living in Athens, such as Herodotus and Gorgias. However, despite being confined to a definite historical period (the second half of the fifth century BCE), the beginnings, fortunes and decline of antilogy as a genre deserve to be investigated further, and certainly not just in order to ascertain whether one could go beyond these geographical and temporal limits.[26]

It is worth adding, in this context, that two groups of thinkers seem to have remained unaffected by the trend toward antilogy: on one hand, 'philosophers' such as Anaxagoras, Diogenes of Apollonia, Philolaos, Melissus, Democritus and Cratylus, and, on the other, 'sophists' and politicians such as Hippias, Critias and Thrasymachus. However, none of them seem to have attained, in Athens, the authority and notoriety of the dozen or so authors that have been discussed above.

## 3. The Significance and Cultural Context of Antilogies

### 3.1. The Novelty of Antilogies

Having outlined an inventory of the antilogies that go back to fifth-century Athens and are still available, it is time to ask how significant the efflorescence of this literary genre is likely to have been, and how an overall cultural context possibly supported such an innovation. For if so many antilogies were written during a few decades—not before, not after, and, as it seems, only or mostly in Athens—it is reasonable to imagine that there was quite a favorable context for the sudden emergence of this phenomenon.

To begin with, antilogies affected rather directly the basic scheme of the interaction between speakers and audiences. Traditionally, audiences knew and respected the tacit rule that everybody should listen in silence to the poet singing, the choir dancing and singing, the *sophos* teaching, the actors acting, the speakers delivering their speeches either in the *dikastēria* or the great assemblies (in Athens, usually the *Ekklēsia* and the *Boulē*), and possibly the historians portraying past events. This was a widespread practice, deeply rooted in the past, and seemingly related to the assumption that people could only appreciate hearing, attending in silence and, at least with songs, possibly memorizing some lines. With the emergence of antilogy, however, this long-established convention was clearly replaced by a different idea of the intellectual game. Recall, for example, what happened with Gorgias. His *Encomium of Helen* was clearly marked by a good measure of intentional one-sidedness. While it is true that his speech was conceived as a rejoinder to an accusation speech, its failure to assign any agency or responsibility to Helen makes the speech unreliable. However, what happens in his *PTMO* is even more self-defeating, since Gorgias here dares to reject some basic assumptions about being (something exists), knowledge (something is knowable), and communication (something can be communicated). It follows that his *PTMO* cannot reflect the opinions actually held by Gorgias on these topics. What *PTMO* offers is marked by a patent and totally implausible one-sidedness that deeply affects the possibility for this work to offer reliable evidence about Gorgias' ideas.

The question thus arises: Why do one's best *in order* to be judged clearly *un*reliable? The answer seems clear. Gorgias did not want his claims to be taken at face value, that is, as documents informing us about his personal tenets. He could only expect these writings to be taken as provocations, as invitations to re-examine the subject matter and try to outline a personal opinion on the topic *in reaction* to his travels into the improbable. He likely hoped that almost everybody would perceive not only the one-sidedness of his speeches, but also how demanding it could be to mount a personal, independent, and contrarian point of view. We thus begin to recognize the essential purpose of Gorgias' intellectual

provocations: to launch a sort of challenge to his audience(s), not without encouraging them to react one way or the other.

The antilogies were indeed conceived of as intellectual exercises and provocations to think—perhaps even jokes—requesting from the audience, first, that the provocation not escape their notice; second, that they enjoy it; and third, that they accept the challenge to raise objections to the author's more or less bizarre claims. Clearly, not every antilogy was as extreme as the *PTMO*. Since the basic aim was to devise two contrasting claims that were both equally plausible (guilt versus innocence, trustworthiness versus untrustworthiness, or other alternatives), sometimes this was judged enough, but, in principle, the surprise, or the patent indefensibility of a given position, was also part of the antilogical game.

In these conditions, the old tacit rule that auditors should listen in silence was going to be put aside. Instead of being asked to remain silent, audiences were now expected to react one way or the other, perhaps as interlocutors, rather than as spectators of an almost theatrical performance, who were prepared to comment on the spectacle in some way. Probably authors of antilogies bet on the readiness of their audiences to react and join in the intellectual game, much as the dialectician Zeno of Elea bet on the reaction of his audiences when he challenged them with his paradoxical stories.[27] In particular, for the audience this meant that one could feel oneself encouraged to generate opinions of one's own and exchange them informally instead of remaining in silence.

In this way, a completely new idea of spectators (as well as of every other audience and, in the final analysis, readers) emerges, although it remained implicit, and new avenues for novel forms of audience reception and mental processing were opened thanks to the art of antilogy. Indeed, antilogies could only encourage members of the audience to think, reason, reflect, and form ideas on their own, while the primary aim (to defeat the adversary) was becoming secondary, and an unexpected interpretative freedom was granted to the public, much as if they were serving as judges even outside *dikasteria*. Public communication, in turn, is likely to have become more sophisticated, more intense and more bizarre thanks to the central role now ascribed to the audience.[28] Therefore, to find a way to react to antilogies is likely to have been, for the Athenians of the fifth century BCE, a demanding but uncommonly attractive challenge.

### 3.2. Further Features of Antilogies

To ensure that an antilogy was recognized as such, authors did their best to mount couples of speeches that were well balanced in their opposition. The ideal became to mount perfectly symmetrical juxtapositions, and authors therefore strove to eliminate any claims and arguments that might violate the rules of the 'perfect' antilogical game. As a consequence, authors often dared to stress the one-sidedness of each speaker's claims, even to the detriment of reliability. This circumstance suggests a growing interest in texts that went beyond statements that can be taken at face value toward intellectual provocation and intellectual perplexities that force the audience to think. Because of this orientation, antilogies became a sort of new game based on new assumptions and an original kind of entertainment that—despite what they had in common with drama—needed no theatre or actors but at most one or two gifted readers. This level of complexity appears at its best in the works of some *sophoi* rather than in the theatrical texts of the same period. The relative brevity of each speech was also helpful in ensuring that the audience's attention did not wander and that the dramatic contraposition of arguments was absolutely alive. These features, in turn, became a criterion for establishing whether or not a certain discourse was conceived as an antilogy.

Taken together, these developments indicate that a powerful trend affirmed itself in classical Athens, to the extent that few distinguished Athenian intellectuals of the period refrained from authoring antilogies, one way or the other. The same holds true for intellectuals living in Athens but coming from abroad, such as Herodotus and Gorgias. And if we look back, we can only conclude that it took just a century and a half—starting from the times of Thales—for the search for primary ideas about individual natural phenomena[29] to

be temporarily set aside in favor of a completely new idea of what befits an intellectual: not to discover and teach, but rather to put into circulation controversial and often unreliable ideas so as to tease and stimulate the public in quite original ways. It is as if for a while the convictions of individual authors mattered much less than the art of puzzling and provoking the public into thinking.

Therefore, the widespread popularity of such extraordinarily elaborate artifacts suggests that an important change of taste took place and a new[30] idea of excellence emerged, an excellence no longer oriented toward content and cognition—as happened with the sophisticated knowledge of the typical pre-Socratic *sophos*—but toward meta-cognition. Indeed, a new phase began in which rhetorical skill manifested itself in exploiting a given situation in order to concentrate on clues and circumstances (either to be enhanced or neutralized), and which materialized in the form of antilogies, *logoi amarturoi* and dramatic agons.[31] These various modes of antilogy imply a new attention devoted to complexity as well as to the exhibition of high levels of liveliness and readiness in dealing with the most variegated details of a given story.

More precisely, the distance between author and public was maintained, since even the antilogy was inspired by the democratic confrontation of proposals in the assembly, where the *dēmos* would have voted for proposal A, as illustrated by speaker X, or for proposal B, as illustrated by speaker Y, with no real possibility of drafting his proposal C on the spot (not to mention that in court every third way was precluded by definition). The portrayal of the 'others' (and therefore also the idea of the public) that emerges, in particular, from four comedies by Aristophanes (the *Knights*, the *Clouds*, the *Wasps* and the *Ecclesiazusae*) constitutes, if I am not mistaken, a splendid materialization of the idea that the public can only be the defenseless prey of speeches. Indeed, the professionalization of the way of dealing with the public helps, in turn, to understand that this was the moment in which oratory was established as a form of excellence and a profitable art that could be taught and that exhibited its usefulness to the fullest in city assemblies and in the courts, with the parties applying to the masters of the word for a fee, confident in receiving the support they needed in order not to risk too much in a political or legal dispute.

But a second message became apparent when antilogies began to emerge because of the creative and original ideas launched through them by an increasing number of distinguished intellectuals. Another logic was trying to affirm itself, with the public invited to think, grasp, understand, appreciate and comment. It amounted to a shift in taste. Because of this new form of literary judgment, the sort of culture that had affirmed itself in the period from Thales to Zeno probably lost some of its attraction and became rather difficult to understand, because the knowledge that the man of science had (and his primary ideas) was no longer the center of attention: its place was being supplanted by the virtuosic play of the skilled speaker and the emotions he was able to arouse in his audience. The proliferation of antilogies thus inaugurated a new epoch, because the emergence of a qualified—and perhaps vast—audience prepared to think, reason, reflect and form ideas was a unique and powerful historical event.

### 3.3. Why Antilogies Flourished in Athens

The sudden emergence and popularity of antilogies in the fifth century BCE raise a question: Why did they come into being in Athens at this time? Structural innovations, such as those concerning the idea of *sophos* and excellence in rhetoric, argument and persuasion, do not happen by chance. They need favorable circumstances, and we are considering a period when Athens was marked by a number of extraordinary events. In the second half of the fifth century Athens offered, indeed, a unique combination of positive stimuli. How could this happen?

An event occurred elsewhere, however, that deserves to be mentioned first. In 494 BCE, just before the Persian Wars, the city of Miletus was destroyed and its adult population was almost certainly submitted to a severe process of deportation. Whatever the details of this event, which are largely unknown, the glorious cultural heritage Miletus was hosting at

that time—truly avant-garde for the whole Hellenic world—was almost destroyed, and there is little information about distinguished Milesian refugees that found asylum in other Greek *poleis*. Because of that, the intellectual leadership of Miletus and its cultural heritage ran into serious danger, to say the least. Then the Persian military expeditions in continental Greece took place. After these dramatic events, another town affirmed itself as a sort of Panhellenic cultural center: not Samos, Croton or Syracuse, but Athens, a *polis* that was marked by a swirling economic, political and cultural expansion that continued for almost a whole century.

A notable feature of the period was the increasing awareness the Athenian *dēmos* acquired of being entitled to much more than merely symbolic access to power. The citizens of Athens had access to some basic political rights and were thus able to make a whole series of steps toward a democratic society while a clearly imperialist policy toward other *poleis* was asserting itself. This was also the period when figurative arts, such as architecture and vase painting, along with the more conventional arts of painting and sculpture, attained extraordinarily high levels of excellence, culminating in the reconfiguration of the entire Athenian acropolis with a number of spectacular monuments and artworks. Contemporaneously, the Attic theatre attained no less impressive levels of excellence, encouraged by the growing involvement of foreign spectators. Indeed, tragic and comic theatre began to assume Panhellenic importance. Finally, the second half of the fifth century was also marked by a conspicuous increase in the use of papyri and people's familiarity with them. We have, indeed, the unexpected privilege of being reasonably well informed about the development of literacy in ancient Athens. To indicate the increasing familiarity with writing on papyri it may be enough, here, to cite a passage from Aristophanes' *Frogs* (414 BCE), a comedy that presupposes a learned public familiar with previous dramas and dramatic authors. "If your fear is that spectators are too ignorant," observes the Chorus, "and cannot understand the subtleties that you two say, do not worry: things are no longer this way. These are people who have gone to war, each of them has a *biblion* [a written roll of papyrus] [and] understands intelligent ideas [*manthanei ta dexia*]" (1112–1114; trans. mine). Clearly "each of them" is an exaggeration, but "a considerable and increasing, and therefore promising, portion of them" would be a reasonable paraphrase of this statement. Furthermore, it is not by chance that in *this* Athens, more and more young men began to attend school, perhaps even schools comparable to the *Phrontistērion* evoked in Aristophanes' *Clouds*.

Taken together, all of these trends and innovations could only instill pride in the *polis*. For an Athenian, it probably was rather easy to perceive a considerable difference between being Attic and, for example, being Spartan, and not just with regard to legal and political rights. More generally, each innovation opened the door to previously unknown, innovative customs and ways of life, and therefore could only reinforce the perception that to live in Athens was largely advantageous (and advantageous because it was much more open to the future) in comparison with other *poleis* or countries. Something comparable probably also occurred elsewhere (Syracuse comes to mind first of all), but not at the same pace as in Athens. This is very helpful in helping us understand how Athens could affirm itself as the most dynamic, vital and interesting city in the world while the art of writing knew a crucial progress.

## 4. Conclusions: Socrates and the Antilogies

No inventory and appraisal of antilogies in fifth-century Athens can ignore the contribution of Socrates and his pupil Plato. At first sight, no clear connection seems apparent, but let me outline the link while anticipating that, at least in my opinion, it would be wrong to leave this connection completely aside.

Socrates' adulthood coincided with that of Protagoras, Gorgias, Prodicus, Sophocles, Euripides, Thucydides and other luminaries. They lived at the same time, in the same place, and immerged in the same intellectual atmosphere. But Socrates distinguished himself from all of them for having rejected the use of papyri and the written word in favor of impromptu verbal interaction and unstructured conversations. Such a choice could well

appear to be anachronistic or even bizarre in comparison with the penchant of all his colleagues for (and their prominent competence in) writing, [32] but it becomes much more understandable when we consider that oral dialogues, though being not foreign to his distinguished contemporaries, allowed Socrates much higher levels of intellectual ductility as he mounted exchanges with individual interlocutors and raised in them perplexities that were at least comparable to those raised by authors of the best antilogies. As several dialogues by Plato suggest, he probably used to start from his interlocutors' ideas and was thus prepared to face their reactions to his questioning, whatever they might be, so as to find a way to keep firm control of the reins of the conversation, which often led his perplexed interlocutors into contradictions and thus into embarrassment. These features are likely to have become a powerful marker of his difference from the majority of intellectuals, who, for the very first time in the context of a mainly oral culture, devoted most of their time to writing and reading. [33] Although Socrates availed himself of other means (basically, unstructured conversations), he seemingly pursued goals strictly comparable to those pursued by the authors of antilogies: surprise, unavailability of immediate answers, and the need to think. Indeed, his dialogues enacted a continuous and explicit invitation to give opinions, in contexts where not engaging in the exchange and saying nothing at all would have been inconceivable—save that to find out 'good' answers often was almost impossible for his interlocutors. Nevertheless, to be reactive and play the dialogic game could well be taken to be a mark of excellence. This way it becomes understandable why Plato remained rather cold towards antilogies: in comparison with dialogue and dialectical questioning, they made only timid and almost insignificant efforts to elicit a reaction from the audience. Nevertheless, without antilogies, Socrates might have never come into being.

From another point of view, both Socrates and the authors of antilogies shared the same lack of interest in points of doctrine and conclusions to be reached. For all of them, putting minds in motion and unblocking thoughts became an extremely attractive option, more important than imposing a definite direction on the thought processes of other people (audiences, interlocutors, and bystanders). A key confirmation of this penchant for what was intellectually surprising—and therefore for the art of putting minds in movement—surfaces when considering that in this period, while Zeno launched his sophisticated paradoxes, treatises were often (though certainly not always) replaced by writings that failed to offer an easily identifiable, positive teaching. Plato's aporetic dialogues continued in the same vein: each time a situation allows room for questions that become more and more difficult to answer, with no final conclusion in sight. In short, in the fifth century, a new intellectual mood affirmed itself thanks to the antilogies and the personality of Socrates, one that took on new life in his oral dialogues and then in the written dialogues of Plato. In 'philosophy', things changed radically only with Aristotle, when the treatise regained its authority and established a supremacy that, as we know, endured in philosophy and other disciplines for millennia.

**Funding:** This research received no external funding.

**Institutional Review Board Statement:** Not applicable.

**Informed Consent Statement:** Not applicable.

**Conflicts of Interest:** The authors declare no conflict of interest.

## Notes

[1]   In his *Republic* (V 454a1–2), Plato mentions the *dunamis tēs antilogikēs technēs*, but without entering the least reference to the antilogies that were still so attractive when he was a young adult. Another passing reference appears in his *Theaetetus*, 154b–e. A rare study of this is (De Luise and Farinetti 2000). See also below, Section 4.

[2]   Since no author labelled his antilogy an "antilogy," this feature had to be detected from time to time.

[3]   Here and below the word 'sophist' is printed with inverted commas because, as Notomi (2010), Tell (2011), and Ramírez Vidal (2016) have convincingly argued, this word began to have a wide circulation only from the beginnings of the fourth century BCE,

when they were identified this way essentially by the *Dissoi logoi* (on which see Note 30 below), then by Plato. Therefore, during their adulthood the so-called sophists probably remained unaware of this qualification.

4    This is, at least, the conclusion reached in (Giombini 2022, p. 216).

5    On the *logos amarturos* there is but a scanty literature (see Rossetti 2012).

6    The main evidence surfaces in Plato's *Phaedrus*, 272c–273a (cf. 259e–260a).

7    Curious evidence of this fact was supplied, in a completely unintentional way, by Henri Passeron in 1970, when he mounted (in a mimeographed typescript that, unfortunately, seems no longer available) a sophisticated argument to conclude that the question was not in fact insoluble. It seems to have escaped Passeron that the story was constructed *in order to* ensure that it remain insoluble, i.e., to make his (and a few others') efforts futile. Besides, when many other scholars (Kerferd 1981 included) recklessly treated the *Encomium of Helen* and the *PTMO* as basic evidence of Gorgias' genuine philosophical beliefs, the same misleading assumption was at work.

8    Sextus Empiricus (*Adversus mathematicos* II 99) adds a significant detail, namely that the judges threw Corax and Tisias out of the courthouse with the proverbial sentence, ἐκ κακοῦ κόρακος κακὸν ᾠόν, or, "from a bad crow a bad egg."

9    I therefore find it rather surprising that the sources in the *Euathlos* continue to be omitted from sourcebooks about Protagoras (with rare exceptions, such as Capizzi 1955).

10   Let me remind that for several decades now the scholarly community has abandoned the distinction between two Antiphons, one a sophist and the other a rhetorician (see Notomi and Giorgini in this special issue).

11   A sustained monograph on these tetralogies is now finally available: (Giombini 2023).

12   One of these summaries is by Sextus Empiricus in his *Adversus mathematicos* (VII 65–87). The other is found in the *Corpus Aristotelicum*, immediately before the *Metaphysics*, as chapter 5 of *De Melisso, Xenophane et Gorgia*. These summaries complete each other. Thanks to them, we can form a definite idea of the ambitious treatise authored by Gorgias.

13   A quick survey is available in (Beerbohm 1922, p. 59 f).

14   In Rossetti (2022). My whole paragraph on Gorgias owes a lot to this article.

15   A key contribution to the identification of these primary claims is (Tordesillas 1990).

16   I shall leave aside other features of the speech.

17   Rossetti (2017) is devoted to stressing their complementarity.

18   Our relative familiarity with the notions of being and ontology can easily mislead us. In all likelihood Gorgias wanted to surprise everybody.

19   Indeed, a lot of creative (primary) ideas emerge. Among them, the notion of "noetic existence" (Sextus Empiricus, *Adversus mathematicos* VII 67) that resurfaced in Aristotle's *Rhetoric* (II 24, 1402a 57) and then as the modern notion of *Gegenstandtheorie* (in Meinong 1904).

20   The other four "architects of the victory", Herodotus continues, shared the latter's opinion, and Otanes gave up power for himself (and his descendants) to maintain his freedom, while Darius got the nomination thanks to a stratagem devised by one of his shrewd assistants of his.

21   As far as I know, nowhere references to the supply of basic information to the *dikastai* are available, much as if their only source of information were the speeches of prosecutor and defendant. See e.g., (Todd 1993, pp. 125–27).

22   In principle, this line of reasoning is consistent with the Socratic demand for competence.

23   These translations from Antisthenes are mine.

24   Unfortunately, these mentions are left aside by Notomi, Tell, Ramírez Vidal (see endnote 3, above), Maso (2018), and others.

25   Just consider the silence that has fallen on the antilogies since the time of Plato and Isocrates.

26   The same with note 25.

27   A recent survey of Zeno's work is available in (Rossetti (2023)).

28   To assess the novelty of this new relationship between authors and audiences, it is interesting to consider that a comparable desire to provoke the spectators resurfaced in theatre only in the twentieth century.

29   We have a primary idea when it is not the mere modification of an old idea, e.g., when Anaximander claimed that the earth is an immense but limited body that has no trouble staying in balance while the sun and other celestial bodies go around it. It is a rather new notion (see Rossetti 2023, scts. 2.2 and 4.3).

30   New with the proviso that for the moment we do not consider the Homeric model (see Rossetti 2023, sct. 1.7).

31   By dramatic contest we mean, at least here, the verbal confrontation between two characters during a theatrical performance in ancient Athens.

32   In 1903 Hermann Diels found it appropriate to qualify his collection of fragments of the most ancient Greek intellectuals as *Die Fragmente der Vorsokratiker*. As a consequence Socrates was left out of this monumental collection. Even if deceptive, Diels' choice raised no problem for a very long time, and Socrates' rejection of writing was a major contributing cause of this. Therefore, it is not by chance that things began to change only in 2016 with the appearance of the comparable collection of Laks and

Most, where a chapter on Socrates is included for the very first time. On a major turn like this one, see also Notomi (2022); (Rossetti (2022, Section 1).

33   Notomi (2022) stresses this point, perhaps going too far when he claims that Socrates "engaged in dialogue. . . as sophists did" and that his' writings "were subsidiary teaching materials rather than main achievements of their profession and inquiry" (p. 8).

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
