# Peer review of "Antilogies in Ancient Athens: An Inventory and Appraisal"

_humanities, doi:10.3390/h12050106_

Round 1
Reviewer 1 Report
Paper title: “Antilogies in Ancient Athens: An Inventory and Appraisal”
After carefully reading the paper that has been sent to me for review, I would recommend the following:
The author of this paper establishes very clearly the two objectives of his text: to make a record of c. 30 antilogies published during the fifth century BC in Athens and expose their characteristic features and their importance.
Considero que el autor alcanzó sus objetivos
General remarks
The author of this article achieves both objectives. Although the record is not complete, as the author himself points out, it offers a clear idea of the large number and importance of this form of discursive and dialogical communication. To the best of my knowledge, although the antilogies had caught the attention of scholars, the studies had only partially addressed this form of discourse, so the paper under my review does a comprehensive exploration of this phenomenon. In addition, the author satisfactorily draws attention to the characteristics of time, authors, place, mode, and purpose of these forms of communication and the impact they had on the culture of the V century BC. in Athens.
So, the paper is novel and significant in classical philology, since it focuses on a much broader set of texts than has been done up to now, but for this very reason it poses new problems, first, the very definition of the term 'antilogy', which seems quite restrictive and does not cover the diversity of texts (cf. comment to “Introductory Remarks”). Likewise, it is necessary to modify the idea that one has of the very nature of antilogies not as a discursive genre, but as a form of communication that can be found in very diverse discursive genres, first, the forensic genre, but also ordinary dialogue and many other types of texts. On the other hand, in terms of temporal and spatial limits, this discursive form cannot be limited to the Athens of the second half of the fifth century, since these manifestations continue to appear, for example, in other texts, such as the Apology of Socrates or the Encomium of Helen by Isocrates, which is also set against other speeches it refutes (perhaps Gorgias's own Encomium of Helen) or the real or fictitious speeches of Attic orators. If this is not the case, then the concept of 'antilogy' should be further restricted or limited.
If these observations are not pertinent, the response to them would mean a further specification and delimitation of this valuable work.
On the other hand, although the author is obviously an expert in the texts he presents, some bibliographical references are missing that allow us to delve into the subject, both sources and studies. For example, in the case of Antisthenes, it would be necessary to point out some works that would allow a better understanding of the brief fictitious speeches of Ayax and Odysseus.
The author must consider these general observations and the ones, but he will decide if he considers them pertinent.
Particular observations.
1. Introductory Remarks
Some introductory notes follow where he establishes the origin of the antilogies (opposite pairs of discourses or contradictory arguments), in Syracuse and Abdera; its flourishing in Athens in the second half of the fifth century and the silence of scholars.
Comment:
He defines antilogies as “sets of contradictory arguments” and then as “pairs of opposed speeches”, or “important types of argument”.
In English, ‘argument’ can have many meanings: argument, plea, debate, quarrel, etc., but not speeches, so a note should be made about the meaning given to "argument". In my opinion, in rhetoric, it is not about arguments, but about speeches or at most 'arguments' (development of reasoning). On the other hand, in dialectic, it is appropriate to use 'argument’ since it is about questions and answers.
The last paragraph of point 1 (“However … particular art”) is repetitive.
2. A Tentative Inventory of Antilogies
2.1. Antilogies by ‘Sophists’
Tisias
De Corax-Tisias offers the example of a trial between a strong and cowardly man and a weak and brave man.
Comment:
The passage “Now suppose, conversely, […]”, introduces a supposition of the Author, not of Plato, who suppresses it with the intention of ridiculing Tisias. So, strictly speaking, this would not be an antilogy.
Protagoras
It refers to the famous trial between Protagoras and Evathlus, first presenting the sources, then exposing the event and finally offering some observations.
Comment:
Regarding the sources, it remains to add DK 80 A1.54 and 56 (= DL IX 54 and 56), and DK 80 B6 (= Quint. Inst. III 1.10).
There is a dispute about that story. For the A., “Protagoras is the most plausible candidate to be the author”. For DK this title is doubtful (Zweifelhafte Titel). Schiappa, Protagoras and logos, p. 215 (not in bibliography), considers that the original lawsuit was between Corax and Tisias, not between Protagoras and Evathlus. In any case, Protagoras is not the author of the story, but a character in the story.
Antiphon
The A. deals meticulously with the three Tetralogies of Antiphon. He first makes a presentation of them as antilogies, pointing out that they are logoi amartyroi, a kind of judicial discourse without witnesses; subsequently he analyzes each Tetralogia and describes the characteristics of those works. For example, in relation to the first, he briefly describes the case, where he highlights the opposition between eikós and eikoteron [these terms are mine] and points out its particularities, for example, the little importance of the alibi argument: “attitude of distrust towards his alibi (a modern notion)”; instead, he points out the perplexity in which the defendant finds himself. Similarly, in Tetr. I: “sense of bewilderment of the defendant”; Tetr. II: “What I have just reported should be enough to convey the spectacular and unpredictable movement from speech to speech in this antilogy”; Tetr. III: “the antilogy makes issuing a verdict under these conditions deeply problematic”. Indeed, it is about what is currently known as "difficult cases", to which the logos amartyros belongs.
Comment:
The Antiphon Tetralogies are not "pairs of opposed speeches", but a set of four fictitious speeches of a judicial nature: two opening speeches and two closing speeches. They reflect the judicial process of the second half of the fifth century in Athens. The problem is that, if we include the Tetralogies in the set of antilogies, we must include the judicial speeches of ancient Greece before and after it, or else consider that the antilogies are only those fictitious speeches.
The testimony is the evidence par excellence in the juries of classical Athens. When these are lacking, we find ourselves in “difficult cases”.
Prodicus
Following the same model, the A. presents the famous theme of Heracles at the Crossroad. First, the A. refers to the source (Xenophon), which offers a summary of that theme that was part of a larger work, the Horai. Next, A. makes a summary of the summary and at the end calls attention to the particularities of this work. In this case, A. draws attention to the difficulty of considering this work an antilogy, since it is a unique discourse, but the antilogy is found in the speeches of the two female characters: Virtue and Vice.
Comment:
Here we are dealing with fictitious speeches within the chapter of a work. They do not belong to the judicial genre like all the previous ones, but to the ethical introspection of the recipient, who is the young disciple of Prodicus. The fact that it is not indicated that the main character decides, does not imply that he must necessarily choose Virtue and not Vice, not the extremes but a middle way. The case is similar to Plato's aporetic dialogues.
Gorgias
In the case of Gorgias, it refers to the three famous works: Encomium of Helen, Defense of Palamedes and On Not Being, or On Nature, speeches that are in opposition to three other speeches, albeit implicitly.
In the case of Helena's Encomium, the Author seeks to identify the means that move the forces that condition the will of the listener and that are beyond his control: gods, fate, persuasion, or love, “almost as if he were writing a treatise on the limits of the will, which is designed to counter the traditional accusation that Helen was the sole cause of the Trojan War”, in such a way that he considers this work as “the first treatise on the limits of the will in Western culture”. The Defense of Palamedes is a fictitious apology in response to Odysseus' accusation, which can clearly be assumed. The author describes the argumentative structure of the work, highlighting the resources used: the possibility, the will and the contradiction of the accusing party, as well as other extra causam evidence. About On Not Being, or On Nature, the A. offers an antilogical interpretation, that is, he identifies the antilogos in common sense, which he tries to refute.
Comment:
Now it is not about a couple of discourses, but a single discourse that can be analyzed on the surface and in the intentionality hidden behind the text, and this crypto text has as its antilogos the traditional discourse of Helen's guilt in the Trojan War. In my opinion, since the pair of an antilogy in this case is the written discourse and the general traditional one, it is necessary to redefine the definitions that have already been given, or else qualify them so that they cover this new antilogical form, which would be, rather a form of refutation (apologia, lysis, anti-sillogismus, elenchus, enstasis) that contradicts not a specific discourse, but a thesis or an idea shared by all.
The Defense of Palamedes is also a refutation (“to reach excellence in making Palamedes' self-defense such a superb refutation”), but in this case in relation to an implicit speech, that is, the concrete speech of accusation by Odysseus.
For its part, On Not Being, or On Nature has common sense as its opposite discourse, although it seemed that the book of Melissus was.
With the previous examples it will be possible to observe that antilogy is not a discursive genre but a type or form of discourse that places the accent on the refutative part.
2.2. Antilogies in Historiography
Herodotus
Analyze the “pairs of antithetical speeches” in Herodotus and Thucydides, first, the famous one, “the celebrated tripolitikos logos (III 80–82) or threefold political speech”, “whose aim was not to teach or persuade but to generate a stubborn perplexity”, as seen in the Antiphon Tetralogies.
Comment:
It should be noted that this is not a couple of contradictory discourses, but three, in which “the drawbacks of just one of the others” are pointed out. In this case, it goes from judicial discourse to political analysis of the forms of government.
Thucydides
Here he analyzes the "Dialogue on the Mytilenians" (III 37-49)", which is not really a dialogue, but a debate before the assembly where two opposing speeches are delivered, the one that was in favor of punishing the Mytilenians with death the Mytilenians and the other who was against: "here the weight of one point of view is perfectly balanced by the other point of view", according to the words of Thucydides himself (III 49). On the other hand, the second example, the "Dialogue of the Melians" is a series of exchanges of points of view where the interlocutors are the ambassadors and the leaders of the island of Melos.
Comment:
The discourse genre of the "Dialogue on the Mytilenians" is deliberative: it is a discourse of exhortation and another of dissuasion, which shows the internalization of discursive strategies manifest in the great political debates during the Peloponnesian War reworked by Thucydides.
In the case of the "Dialogue of the Melians" the antilogy is already an embassy dialogue, where both parties refute each other, so that it is not a pair of opposing discourses, but rather a discursive continuum.
2.3. Some Further Antilogies: Antisthenes’ Ajax and Odysseus, and the Dissoi Logoi
Antisthenes
The autor is limited to analyzing two conflicting speeches: Ayax and Odysseus [published when Antisthenes was not yet a disciple of Socrates]. In Ajax, Antisthenes introduces a series of arguments about his right to possess Achilles' weapons. In the case of the Odysseus speech, the arguments are also analyzed, but they appear in a chaotic way, so the author presupposes that this was because Antisthenes was very young when he wrote that speech. The A. reaches the following conclusion: “Because of this lack of order, the present antilogy could be an early work, one endowed with originality and force but weakened by an evident compositional flaw”.
Comment:
It is not mentioned that Antisthenes was first a disciple of Gorgias, before joining Socrates, and that his education was therefore first in rhetoric and later in dialectic. Both speeches are very short, but they are clearly related to the Gorgias speeches discussed above, so they could be a product of that school.
The fact that they are his first works does not imply that they had defects in order.
If the Ajax and the Odysseus make up an antilogy, then they can be related to some works from the author's Socratic period, where he theoretically approaches that discursive form. That is, there is a correspondence and sequence between the works of the Gorgian and Socratic periods.
The Dissoi Logoi
As in other sections, A. makes a presentation of these anonymous works, then describes them and at the end makes some pertinent observations.
Comment:
At the end it refers to the use of the term sophistês in these works, which would be important to date the work, although no hypothesis is offered about its chronology.
2.4. Antilogies in Tragic and Comic Theatre
Sophocles
A. observes that Aeschylus' tragedies could have included antilogical discourses, but this did not happen, so he concludes that “The antilogies probably remained unknown to Aeschylus”. On the other hand, Sophocles introduces antilogies (“clash of opposed speeches”) in his tragedies, which the A. analyzes in a general way, drawing attention to characteristics that allow us to think that Sophocles knew about antilogies and knew how they worked, although it is doubtful that he used them intentionally.
Euripides
Instead, Euripides consciously introduces stupendous antilogies into his tragedies. A. analyzes a direct verbal confrontation between Hecuba and Helen in The Trojan Women. Other tragedies include noteworthy contrasting speeches. The author of tragedies had received the influence of this mode of discursive presentation from the 'sophists'.
Comment:
There seems to be an evolution in the use of antilogies in Greek tragedy that goes from their non-existence in Aeschylus and from their perhaps unconscious use in Sophocles to a formidable use of these discursive forms in Euripides. The use of this discursive form, in my opinion, is not the work of some teachers, but was due to the diffusion and internalization of certain discursive forms that became increasingly frequent and pregnant due to political and social changes that occurred in Athens throughout the 5th century B.C.
Aristophanes
In Aristophanes these discursive novelties are manifested. In the Clouds the verbal exchange between fair and unfair speeches is shown, but the A. observes that “What we read here is not a pair of speeches but two steps of a sustained contraposition”. The A. describes examples of antilogies in other comedies and comes to the following conclusion: “[…] a more systematic investigation […] show that, at least in the last three decades of the fifth century BCE, the most celebrated playwrights showed that they were quite familiar with the new literary genre of antilogy […] Indeed, the plays offer the best proof of how antilogy permeated the Athenian avant-garde of the period and indicate the success of antilogy as a new form of argument and composition”.
Comment:
It will be possible to notice that the antilogies in the comedy appear either in the form of opposed speeches or in sustained dialogues between two characters that can be mutually interrupted. It is probable that antilogies arose first in the legal sphere and then spread to other forms of communication such as dialogical exchanges.
2.5. Some Basic Remarks on this Inventory
The A. exposes some observations of the study carried out. Among other aspects, he points out that "Moreover, with just one or two exceptions, all of these writers worked in Athens in the second half of the fifth century BCE—none before, none after, none else", so it should be considered a feature characteristic of the time. Furthermore, the number of examples of this form of communication must have been much larger than those included in this study. There were, however, several writers who seem not to have been influenced by this form of communication.
3. The Significance and Cultural Context of Antilogies
3.1. The Novelty of Antilogies
3.2. Further Features of Antilogies
3.3. Why Antilogies Flourished in Athens
A series of important observations are presented about the changes that were generated with the diffusion of this mode of communication, such as the emergence of new receivers, a public that was prompted to think and reflect differently from how it had been done until then. . It was a novel exercise of the intellect.
Comment:
Perhaps it would be important to note that antilogy is not a genre, but a form of communication that appears in all genres, whether discursive (particularly in the judicial genre) or dialogic.
When one says "none before, none after", the contrasting speeches pronounced in the juries and assemblies of the fourth century are not considered antilogies.
As for the public, the recipients are not only the spectators, but also the judges and the disciples who listened to the master's lessons.
4. Conclusion: Socrates and the Antilogies
The Socratic dialogue is another mode of reasoning process, although the author points out some similarities with antilogies.
Comment
However, it has been seen that in comedy the dialogic exchanges followed the model of antilogies, which was favored in the second half of the fifth century, but which already existed (e.g. the Hymn to Hermes) and would continue afterwards, although it would no longer be characteristic of the time. In the Athenian courts of the fourth century the process was still like that of the previous century.
References
Falta:
SSR
Reames 2022
NA
Author Response
GENERAL REMARKS True that, in a sense, all forensic speeches and Plato's Apology are antilogies, but they are not meant to be antilogies. Secondly, Isocrates' Helen probably is just a late fruit, but I agree that to enter an ad hoc paragraph on Isocrates would have been a good idea. Thirdly, the speeches of Antisthenes have received so far only a very very limited attention. Arguments vs. speeches: arguments form the essence of speeches, they are strictly related and terms are interchangeable, as the reviewer him/herself does. 2.1.Tisias To point out that the sentence "now suppose etc." expands Plato wouldn't be a sort of pedantry? 2.1. Protagoras To list all the relevant sources and all the relevant bibliography was not part of my tasks. 2.1. Antiphon true, but is the reference to 'pairs' really misleading? 2.1. Prodicus With respect: (A) Gorgias' Helen is equally fictitious; (B) the story of Hersvcles is the story of somebody that chose Arete, though in a rather special sense. No intermediate option was considered. 2.1. Gorgias I don't like definitions and chose to adopt a rather flexible idea of Antilogy. 2.2. Herodotus If the options are three instead of two, but each has just one target, should we mount an ad hoc re-definition of antilogy? I suppose we shouldn't. 2.2. Thucydides I have the same perplexities 2.3. Antisthenes Sorry, I dare to disagree. 2.3. Dissoi logoi Chronology would be out of the range of interests I tried to cover here. 2.4. Sophocles, Euripides I accept with deep gratitude the reviewer's remarks. Indeed, things could have happened in the way he/she suggests. 2.4. Aristophanes Nothing to say 3.3. Why flourished I argued above why judicial speeches are not exactly antilogies: because they adhere to the facts, rather than to the antilogical scheme. 4. Conclusion It is not clear where the reviewer's remarks lead.Reviewer 2 Report
The article deals with a relevant and significant topic for classical studies. It reviews the main cases of antilogies and proposes in several cases concise but interesting interpretations of the passages. The attempt to bring together the main sources and connect them within a general framework gives it value and can be very useful in drawing attention to the importance of the subject and in introducing the problem. A few other cases could be added, but the selection is appropriate and sufficient to support the central idea. The structure is clear and connects the arguments well. For this reason, I recommend its publication.
Author Response
I can only offer my warm thanksgiving.